# Trolley dilemma in the sky: Context matters when civilians and cadets make remotely piloted aircraft decisions

Markus Christen[1]*, Darcia Narvaez[2], Julaine D. Zenk[2], Michael Villano[2], Charles R. Crowell[2], Daniel R. Moore[3]

1 Digital Society Initiative & Institute of Biomedical Ethics and History of Medicine, University of Zurich, Zürich, Switzerland, 2 Department of Psychology, University of Notre Dame, South Bend, Indiana, United States of America, 3 United States Air Force Academy, Colorado Springs, Colorado, United States of America

* christen@ethik.uzh.ch

**Data Availability Statement:** Data is available in Zenodo: (https://doi.org/10.5281/zenodo.4519817).

## Abstract

Crews operating remotely piloted aircrafts (RPAs) in military operations may be among the few that truly experience tragic dilemmas similar to the famous Trolley Problem. In order to analyze decision-making and emotional conflict of RPA operators within Trolley-Problem-like dilemma situations, we created an RPA simulation that varied mission contexts (firefighter, military and surveillance as a control condition) and the social "value" of a potential victim. We found that participants (Air Force cadets and civilian students) were less likely to make the common utilitarian choice (sacrificing one to save five), when the value of the one increased, especially in the military context. However, in the firefighter context, this decision pattern was much less pronounced. The results demonstrate behavioral and justification differences when people are more invested in a particular context despite ostensibly similar dilemmas.

## Introduction

The Trolley Problem [1] has become a cornerstone in moral decision-making research. This paradigm is a hypothetical dilemma involving, in its original version, a choice of either letting a runaway trolley kill five victims on a track, or redirecting the trolley, which results in sacrificing one victim on a sidetrack. The Trolley Problem has been modified in various ways to assess the effects of physical directness of killing, personal risk to the subject, inevitability of the death, and intentionality of the action on Trolley Problem decision-making [2]. Furthermore, it has been used in various methodological and application contexts, for example to study neuronal responses to such dilemmas [3], framing effects of these dilemmas [4, 5], and autonomous driving [6, 7]. Typically, trade-off decisions are guided by a cost/benefit analysis in which the most positive (or least negative) outcome is selected for the greater good [8]. The "redirect decision" to save the five over the one by redirecting the trolley is often viewed as the "utilitarian choice" which is the decision most people make in the original Trolley Problem [9, 10]. However, if the degree of personal involvement of the actor [11] or characteristics of

**Funding:** This project has been supported in part by grants from the US Air Force Office of Scientific Research (award number 16RT0881) and by armasuisse Science and Technology (S+T), via the Swiss Center for Drones and Robotics (SDRZ) of the Department of Defense, Civil Protection and Sport (DDPS).

**Competing interests:** The authors have declared that no competing interests exist.

potential victims, such as genetic [12] or familial [13] relatedness, are manipulated, decision patterns change as stronger emotional conflict diminishes the likelihood of the utilitarian choice (we note that this does not necessarily imply a dual-process model that clearly distinguishes between intuition versus deliberation, as recent research has shown that emotions play a complex role in utilitarian moral judgment [14]). For example, referring to the Moral Foundations Theory [15], other studies have shown that relational context substantially influences third-party judgment of moral violations, and explains variability in moral judgment independent of several factors that have consistently been shown to strongly correlate with moral judgment (e.g., political ideology; moral value endorsements).

Moreover, these diverse approaches using the Trolley Problem in moral research are built on inferences from participants' responses, in most cases, to decontextualized, impoverished stimuli (i.e. paper-and-pencil tasks). Additionally, moral psychologists have examined evaluations of poorly guarded trolleys, strangers with odd sexual proclivities, and endorsement of abstract principles [16]. Only a few studies have systematically analyzed whether the context of the decision problem matters as well. A particularly relevant context to vary in that respect is war versus peace, as tragic choices are not an extremely rare occurrence in war-like settings. One study that used variants of the Trolley Problem (but still remained in a paper-and-pencil methodology) indeed found that third-party observers judge a trade-off of one life for five as more morally acceptable in war than in peace, especially if the one person is from an outgroup of the person making the trade-off [17].

The rarity of context-related studies indeed underscores the lack of investigation of this factor as a relevant shortcoming given the critique that the Trolley Problem is a highly unrealistic, or uncommon, scenario [18–21]. Furthermore, the common paper-and-pencil surveys involving conceptual scenarios limit engagement of decision-makers. Recent studies using 3-D virtual reality [22] or computer simulations [23] surprisingly demonstrate that more realistic dilemmas enhance the utilitarian choice [24], despite invoking greater emotional responses in the decision-maker—an outcome shown in other studies to be associated with fewer utilitarian choices [25]. These findings point to the well-known problem that judging and acting are not the same [26, 27] and make the role of emotions in Trolley Problem decision-making puzzling.

In order to advance the research on contextualizing "trolleyology," we applied the Trolley Problem to a decision context where tragic choices are not an extremely rare occurrence. Further, we used a methodology where decision-makers had realistic agency through their guidance of systems with harming capabilities. Military remotely piloted aircraft (RPA) operators often make difficult decisions about whether or not to inflict harm on individuals or groups [28, 29]. These decisions may involve a trolley-like moral dilemma where the RPA operator and crew must choose between (a) killing notorious terrorists and sacrificing nearby innocent bystanders, or (b) doing nothing, which will spare bystanders but enable terrorists to commit future acts of violence. We note that this dilemma is not exactly equivalent to the Trolley Problem, as it involved another layer of uncertainty given that the potential harm that the terrorists can cause is in the future, which involves some uncertainty as to whether or not the future harm actually will take place. Despite this difference, using the Trolley Problem as a framework creates a jumping off point to investigate these dilemmas through a well-established moral decision-making paradigm. In this way, the ethical, legal, and strategic implications of weaponized RPAs [30, 31] can be investigated through trolley-like decision-making in an RPA context that has rarely been studied (for an example see [32]).

In our studies, we utilized the RPA setting but adapted the "terrorist" vs. "bystander" dilemma to better match the original Trolley Problem. In our dilemma, the decision-maker was confronted with a "friendly fire" dilemma, as the identified target turned out to be friendly

troops (in the military condition). In a civilian scenario-condition, firefighters were in harm's way of an action meant to extinguish a forest fire. In both cases, the five identified causality deaths could only be prevented by intentionally killing an individual at an alternate location. Our interest was to see how the context of a dilemma affected decision patterns, emotionality, and reasoning.

Using a between-subject design, we contrasted three structurally similar dilemmas: military use of RPAs (intention to harm; military condition) with civilian uses of RPAs (intention to help; firefighting condition), or surveillance (benign intention; control condition). Using an RPA simulation (see Fig 1 and Materials and Methods), subjects within each of the three between-subjects dilemma conditions experienced a within-subjects manipulation involving three trolley-like dilemma "missions" with escalating value of the potential single victim (Peer, Commander, Family Member) across the missions. The experiment also included pre- and post-tests (see Materials and Methods). Stronger emotional conflict was measured and signified by longer decision times (DT) during decision-making tasks [3, 25]. Reasoning associated with the dilemma choices was also measured through a hypothetical reasons measure and post-study explanations of simulation choices (see Materials and Methods for details). Finally, because population characteristics can influence moral decision-making [33], we included samples from both civilian undergraduates and cadets at the United States Air Force Academy (USAFA).

### Hypotheses

The considerations above led to three hypotheses that were investigated in our study (see Table 1 for an overview). First, consistent with previous findings [11–13, 15], we hypothesized that escalation of the "personal" nature of the dilemma across missions would reduce utilitarian choices (save five) in favor of the deontological alternative (save one). Second, in accord with previous findings [3, 25, 34, 35], we hypothesized that signs of emotional conflict as indicated by longer reaction times would be more evident in the Commander and Family Member missions compared to the Peer missions. We expected that this hypothesis would hold for both moral dilemmas (i.e., military and firefighter), but not surveillance. We did not expect significant differences in response patterns and emotionality when comparing the military and firefighter conditions. Instead, we expected *population differences* such that the utilitarian response pattern would be enhanced for USAFA cadets when compared to civilian undergraduates in the military Peer mission, because cadets might value the lives of soldiers more than typical college students [36]. However, we expected to have a greater separation between Peer and Commander missions for cadets than for civilian students because of cadets' respect for authority.

Third, we were also interested in differences between participants electing to redirect versus those who did not (the two samples were not combined for this analysis). "Non-redirectors" were defined as those participants who refrained from the utilitarian choice in at least two out of three dilemma missions; the remaining participants were "redirectors." In line with previous findings [4, 8, 10, 11, 37], we hypothesized that "redirectors" would show more attachment to abstract philosophical reasons to justify their actions compared to the "non-redirectors." Again, we expected that this would hold only for both dilemma conditions; whereas, for the surveillance condition, we expected justifications to be largely non-philosophical. No differences in this tendency were expected across populations.

## Materials and methods

### Participants

There were two different samples. Sample 1 included 172 civilian, undergraduates from a Midwestern private university ($M_{age}$ = 19.1 years; 58% female) who volunteered through SONA

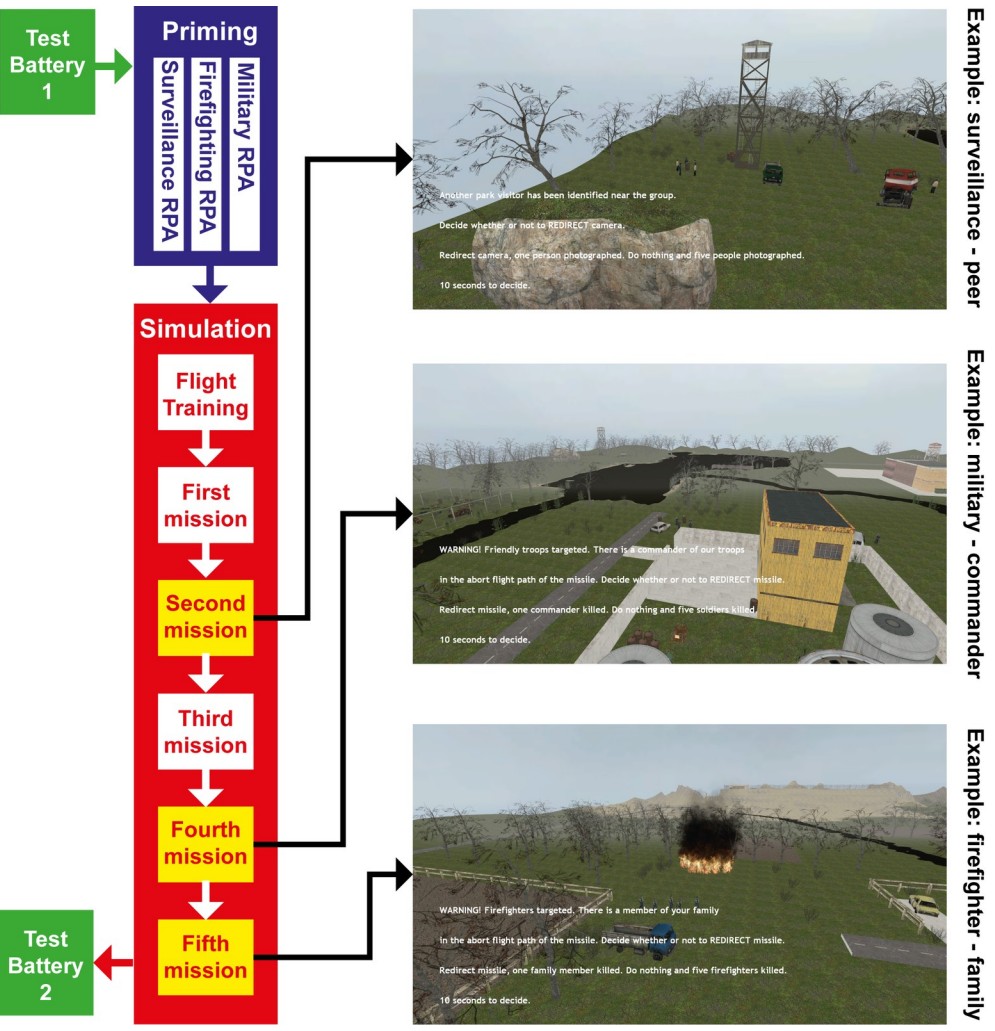

**Fig 1. Outline of the RPA simulation.** Left side: Experiment's design. Right side: RPA simulation. Top: First choice–five versus one peer (surveillance; control condition). Middle: Second choice–soldiers vs. commander (military context). Bottom: Third choice–firefighters vs. family member (firefighter condition).

**Table 1. Overview of hypotheses and measures.**

| General Hypotheses | Samples Differences | Measure |
|---|---|---|
| Escalation of the "personal" nature of the dilemma across missions would reduce utilitarian choices (save five) in favor of the deontological alternative (save one). | No significant differences between samples overall, but possible differences within Military scenario, especially with the Commander mission | Choices made |
| Signs of emotional conflict as indicated by longer decision times would be more evident in the Commander and Family Member missions compared to the Peer missions | Less evidence of increasing DT in USAFA population due to training and background. | Decision time |
| "Redirectors" would show more attachment to abstract philosophical reasons to justify their actions compared to the "non-redirectors." | No significant difference between samples | Reason measure |

Systems, an online participant management and recruitment software, and were offered course credit or monetary compensation for their time. They were assigned randomly to one of the three conditions: Military (n = 53), Firefighting (n = 52), and Surveillance (n = 67). Within this sample, there were two Reserve Officer Training Corps (ROTC) cadets.

Sample 2 included 75 USAFA undergraduate cadets ($M_{age}$ = 19.9; 23% female) who volunteered for the study through USAFA's SONA System. They were offered extra credit towards their course grades. They also were randomly assigned to conditions: Military (n = 22), Firefighting (n = 26), and Surveillance (n = 27). Within the USAFA population, there were two Reserve Officer Training Corps (ROTC) cadet exchange students.

All studies have been performed following the ethics requirement of the involved institutions; all participants provided informed consent. The study has been approved both by the Institutional Human Research Protections Program of the University of Notre Dame (details available on: https://research.nd.edu/assets/187718/institutional_human_research_protections_program_policy.pdf) and by the USAFA Human Research Protection Program (details available on: https://www.usafa.af.mil/Portals/21/documents/Leadership/PlansAndPrograms/IRB/USAFAPoliciesandProcedures.pdf?ver=2016-08-08-164542-397).

## Experimental design

The study was a controlled laboratory study with a between-group design comparing across three different scenarios and two populations. The outline of the experiment is described in Fig 1. Research objective and hypotheses are outlined in the introduction and in Table 1.

Below, we briefly explain the main elements of our experimental design:

- *Sample size*: For this study, we had no a priori knowledge of what effect size to expect, which is a crucial consideration in determining statistical power [38]. Moreover, due to the voluntary nature of subject recruitment from our subject pools, it was difficult to predict the number of subjects that would elect to participate. Accordingly, using the guidelines provided by Cohen [38], we elected a minimum sample size per group (N = 30) in each population needed to detect mean differences with a power of 0.8 and an alpha of 0.05 (52 subjects per group are needed to achieve a power of 0.8, with medium effect size and alpha = 0.05. We had 55–67 per group for the three scenarios. Not knowing what effect size to expect, we elected medium as a middle of the road between large and small effect sizes, see also the paragraph on analysis strategy in the result section). We were able to meet or exceed this target number in both populations.

- *Rules for stopping data collection*: No specific rules for stopping data collection were in place.

- *Data inclusion/exclusion criteria*: Persons where technical errors in the simulation occurred (26 cases) or that failed to perform the simulation properly (11 cases) were excluded from the study. Four persons that registered for the study did not show up.

- *Outliers*: No outliers have been identified in the analysis; no outlier was excluded.

- *Selection of endpoints*: Measures used in the study are outlined below. Primary endpoints in the sense of a clinical study were not prospectively selected.

- *Replicates*: The experiment was repeated once for each of the two populations studied using the same IVs and experimental parameters. Each subject in each population performed the experiment once.

- *Randomization*: Subjects in each replication were randomly assigned to the three scenarios.

• *Blinding*: Only the participants were blinded with respect to the scenario in which the simulation took place.

Further details regarding scenarios, simulation, research subjects and measurement techniques used are outlined below.

## Scenario description

**Military/Combat scenario.**    The backstory for this scenario was a country involved in a war against terror. The RPA pilot's objective was to find groups of terrorists using directions given by the "Central Command" through radio messages in the simulation that could be heard as well as read on screen. For every mission, when the RPA reached a defined target location, a missile was fired towards a group of five persons who were suspected terrorists. However, in some of these missions, after the missile was fired, the pilot was informed that the original target group was misidentified by intelligence as terrorists and were actually friendly soldiers. The choices available to the pilot at that point were to do nothing, which would kill the five friendly soldiers, or to redirect the missile to another target within ten seconds, which would result in the death of one person who was at the redirected location. Thus, this scenario involved a dilemma patterned after the Trolley Problem such that the choices were to save the five and sacrifice the one (redirect decision) or save the one and sacrifice the five (non-redirect decision).

**Firefighting/Disaster management scenario.**    The backstory for this scenario was a country threatened by large forest fires. The RPA pilot's objective was to navigate to high intensity fires that needed to be hit by water missiles to extinguish the fire. For some missions, five firemen suddenly appeared on scene and would be hit by the missile, if no decision is made to redirect it. As in the military scenario, redirecting the missile resulted in the death of one person but saved five.

**Surveillance scenario (pseudo-dilemma control condition).**    The backstory for this scenario was a census that needed to be conducted in a national park. The RPA pilot's mission was to find groups of persons in the park and photograph them. The pseudo-dilemma choice presented here was to photograph five persons or redirect the camera to photograph only one. This control condition was intended to de-confound action vs. moral code [39].

## Simulator description

The first sample was collected at a private, Midwestern university, the civilian site. The experiment took place in an acoustic booth containing a large 60-inch LED monitor. The booth was utilized to ensure immersion into the simulation scenarios. Participants also wore headsets with microphones in order to stay in contact with the experimenter while receiving prerecorded mission orders and responding, as needed. A joystick was used to operate the RPA during the simulation, which was exchanged for a mouse for the purpose of completing the pre- and post-simulation scales.

A measure of pulse rate was obtained from a finger pulse-ox monitor attached to the index finger of the participant's non-dominant hand, which was slightly restrained to minimize movement artifacts. The reporting of the heart rate data is beyond the scope of this paper.

The second sample was collected at USAFA, the study took place in a flight simulator featuring an 80-inch high-fidelity monitor to ensure immersion for the simulation scenarios. The same configuration was used to contact experimenters, in controlling the simulation, and to measure pulse rate throughout the simulation as was used for the first sample.

The RPA simulation was created by one of the authors utilizing Valve Software's Half-Life 2 Software Development Kit [40]. The Hammer Editor was used to create the game level or map

that included a base from which the RPA took off and landed as well as other features, such as a car junk yard, a water tower, a factory, a barn area and a desert with ruins. Since Half-Life 2 is a first-person point of view video game, flight was simulated through the creation of an invisible ramp and an extensive invisible "floor" constructed above the terrain. Thus, in controlling the "player" with a joystick, the camera moved as though it was a camera on the underside of an RPA giving an effective illusion of a flight camera. Invisible triggers implemented throughout the map were utilized to give consistent commands of how to proceed as both visual (text on screen) and auditory (recorded text-to-speech sound file) prompts. In addition, the game source code was modified to enable data collection regarding game events and participant decisions and reaction times, which were written into a plain text log file.

In the simulation, the participant used a joystick to "fly" through the 3D environment while taking on the role of an RPA pilot, guiding the RPA from launch to landing while completing exploration, training, and mission phases. During the simulation, the participants did not decide whether the target was legitimate nor did they launch the simulated weapon; however, they were able to shift the missile target to an alternate path by pressing the joystick's top button, which resulted in some amount of collateral damage for the missions with embedded Trolley problems. During missile control, camera controls were automated to follow the missile's flight path to detonation. Once the missile had detonated, camera view and movement control were returned to the participant. After impact, the pilots had to approach the scene and report the resulting damage. This target verification feature of the simulation was similar to the actions taken by real RPA operators. All commands and prompts were given by a simulated "Central Command" using both visual and audio modalities.

All scenarios contained five missile-launch events, termed missions. Missions one and three served as baseline sessions and did not confront the participants with any decisions. Missions two, four, and five were dilemmas. Throughout the simulation, each redirect decision and decision time was recorded via the Half-Life SDK simulation. The Surveillance condition presented participants with pseudo-dilemmas for missions two, four and five in which the choice was between photographing one or five persons. As with the other scenarios, the value of the one changed across missions in this condition from a peer, to a park ranger, to a family member.

## Measures–pre-simulation scales

Several measures were administered in the pre-simulation test battery as a way to compare baseline measures for groups (Military, Firefighter, and Surveillance) and populations (Civilians and Cadets). These comparisons served to verify that the different groups involved in this study were comparable before exposure to the simulations on key attitudes and dispositions that were potentially relevant to simulation performance and reactivity. No significant differences were found across most of the baseline measures administered, except as noted below. For the Reflective Imaginative subscale of the Triune Ethics Meta-Theory (TEM) measure a difference was obtained between the Military and Surveillance group ($p = 0.04$) within the first sample. Additionally, differences were obtained between the second sample's Firefighter and Military participants within the Engagement subscale of the TEM ($p = 0.05$). These baseline measures are described below.

**Video game experience.** This measure was based on questions related to the age of participants first video game experience, hours of video games per week currently, overall hours of video games played, experience with 3D video games (such as flight simulations or games taking place in a 3D world), and the percentage of video games which were 3D. The primary measures analyzed here were age of first video game use, hours per week played, and overall hours currently played.

**STAX-2.** The STAX-2 Aggression scale was completed by participants before the simulation to assess group comparability and again after the simulation to assess any possible change in aggression state-trait that might have resulted from simulation exposure. Scores were calculated by summing the various items in this scale into a total score. Higher total scores correlated with higher aggression-states (maximum score 60) while lower scores signified the opposite. The difference between pre- and post-simulation total scores was taken as a measure of change in aggression produced by exposure to each Simulation Scenario.

**Perceived stress reactivity scale.** This pre-simulation measure produced five subscale scores: Prolonged Reactivity, Reactivity to Work Overload, Reactivity to Social Conflict, Reactivity to Failure, and Reactivity to Social Evaluation. Items were summed together (some reverse coded) to produce five subscale scores (maximum scores ranged from 12–15 depending on subscale). From there, the overall score (maximum score 69) was created by summing these five subscale scores. Higher scores across all six scores indicated higher stress reactivity.

**Triune ethics meta-theory (TEM) measure.** The items from this scale were grouped and summed creating four subscale scores for each participant corresponding to four moral orientation types encompassed by the theory: Withdrawal, Opposition, Engagement, and Reflective Imagination [41]. Higher scores (out of 25 for each orientation) signified higher alignment with that particular moral orientation.

## Measures–simulation data

**Mission choices.** Dependent variables here included the choices made in each mission either to Redirect or Not Redirect. For each Scenario (Military, Firefighter, and Surveillance) and Mission (Peer, Commander, Family Member), the proportion of participants making either Redirect or Not Redirect decisions was computed.

**Decision Times (DT).** This measure often has been shown to relate to emotional reactivity related to decision conflict [1, 3, 34, 35, 42] (we note that we do not consider longer DTs as a measure of deliberation following classic dual processing accounts given more recent findings [43], but that our focus is on the decision conflict aspect). DT was computed during the simulations as the time needed to make a decision from the point where the simulation introduced the trolley problem choice ("If you redirect, one soldier will be killed. If you do not redirect, five soldiers will be killed. 10 seconds to decide") to the time when the participant clicked the trigger button on the joystick, enacting the redirect decision. For the analyses using this variable, only redirect decisions were used, as they were the only active choice in our Simulation; non-redirect decisions were always the maximum DT for a particular mission and, therefore, did not vary. In the analysis of DT data, we wanted to evaluate the change in DT across missions. However, since only the DT data for redirectors was used, and because redirectors were not the same for each mission, we elected to treat Missions as a between-subjects factor in an overall DT analysis along with Scenario. This was deemed to be a conservative strategy that would permit an examination of DT changes across missions, especially given the fact that the redirectors in one mission were not necessarily the same participants as the redirectors in another mission. This is further discussed within the analysis below.

## Measures–Post-simulation scales

**Reasoning measure.** The first instrument was a "reasoning" measure using Likert rating scales to determine how participants evaluated arguments that might justify redirect and non-redirect decisions in the original Trolley Problem dilemma. Arguments for both redirect (save the five) and non-redirect (save the one) decisions fell into three categories: Philosophical (e.g., utilitarian or deontological); Inflated victim value (e.g., there is a member of my family among the five, or the one is a member of my family); and Deflated victim value (one of the five has a

terminal illness, or the one has a terminal illness). Each potential justification was evaluated on dimensions reflecting both cognitive and emotional components. The cognitive component assessed how universal and logical the justification was perceived to be, whereas the emotional component assessed how appealing and publicizable the justification was judged to be. We note that although our reason measure does distinguish between those two dimensions (cognitive and emotional), this does not imply that the dimensions are antagonistic; recent research indeed has shown that cognitive and emotional aspects interrelate in complex ways in moral decision making [for a review, see 44] and that interaction effects can occur [14].

Since this reasoning measure was gathered after the experiment, it was employed to determine if scenario condition (Military, Firefighter, Surveillance) had any systematic effects on perceived justifications for non-redirect and redirect decisions. We did not expect justification judgments of participants to differ across Military and Firefighter conditions, but both might differ from Surveillance to the extent that experience with the Military/Firefighter dilemmas sensitized participants to the reasons for their decisions. For example, exposure to either Military or Firefighter conditions might elevate ratings for Philosophical and Inflated victim value dimensions on this test, relative to the Surveillance condition for both populations.

From the various items involved in this scale as described above, a single score was computed for each reason category (Philosophical, Upgrade, Downgrade) within each hypothetical decision (redirect or non-redirect) scenario. In all, there were six summed scores, three (Philosophical, Upgrade, Downgrade) for the hypothetical Redirect and three (Philosophical, Upgrade, Downgrade) for the hypothetical non-redirect scenarios. Each score summed across the two examples and each of the four scales within that reason category (maximum score 40). Cronbach's alpha was calculated for each of the six outcomes and ranged from 0.76 to 0.87.

Through the use of this measure, we wanted to understand how redirectors in this study justified hypothetical redirect decisions, as well as how non-redirectors justified hypothetical non-redirect decisions. Based on participant choices, two grouping variables were created so that Redirectors and Non-Redirectors could be examined separately within each Scenario and Mission (see below, paragraph "Independent variables and analysis strategies"). Using these definitions, we might expect majority redirectors to provide higher ratings for philosophical justifications for redirect decisions, while majority non-redirectors might display higher ratings for more emotional, value-based justifications for non-redirect decisions; such as the Upgraded or Downgraded Victim justifications. It is not clear that any population differences would be expected here.

**Mission review debriefing.** In a final self-report measure, decision-makers were asked about their reasons for the decisions they made during the Peer, Commander and Family Member missions. The reasons provided were categorized as involving either a philosophical (e.g., it is wrong to kill anyone) or non-philosophical (e.g., I couldn't kill my own family member) justification. With this measure, for the Military and Firefighter conditions, we expected to see a progressive shift across missions from largely philosophical justifications in the Peer mission for both redirectors and non-redirectors to largely non-philosophical justifications in the Family Member mission for non-redirectors. However, we believed there would be continued philosophical justifications for redirectors in this mission. For the Surveillance condition, we expected justifications to be largely non-philosophical across all missions. We did not expect differences across populations in this pattern.

## Results

### Independent variables and analysis strategies

Across both population samples in this report (Civilians vs Cadets), our primary objective was to understand the effects of three different real world, modern Trolley Problem scenarios

involving military, firefighting, and surveillance contexts on decision making. Furthermore, multiple dilemmic missions were employed within each scenario to progressively alter the social and emotional value of the one in order to investigate the effects of this factor on choices made. Thus, in both populations, the Trolley Problem Simulation Scenarios (Military, Firefighter, Surveillance) served as a between-subjects independent variable, while Mission (Peer, Commander, Family Member) served as a within-subject, independent variable (see below for an exception). A decision-based grouping factor determined by participant choices to redirect or not (see description below) was used for some analyses as an additional between-subjects factor. Finally, an overarching between-subject independent variable of Population (Civilians vs Cadets) was examined, which allowed for comparisons across the two populations in this study.

Based on participant choices, two grouping variables were created so that Redirectors and Non-Redirectors could be examined separately within each Scenario and Mission. One grouping factor was based on the "majority choices" made by a participant across all three missions. Here, two out of three decisions for a participant defined that person's grouping category such that majority redirectors (MRs) were those participants making at least two R decisions across the three missions while majority non-redirectors (MNRs) were those with at least two NR decisions. A second grouping factor was based on only "unanimous" choices to redirect or not. Unanimous redirectors (URs) were those electing to redirect on each mission, while Unanimous non-redirectors (UNRs) were the converse. These factors were used as an additional between-subjects factors in appropriate analyses below.

Several types of analysis strategies were employed in this study depending on the types of dependent variables considered. The binary (redirect or not) choices made in dilemmic decisions and in the actual reasons measures (philosophical or not) were evaluated in two ways. Between group comparisons of the proportions of choices made across scenarios (Military, Firefighter, and Surveillance) or across populations (Civilians vs Cadets) were assessed using chi-squared tests of independence. Within-group comparisons of these proportions across missions (Peer, Commander, Family Member) were made using Cochran's $Q$ test. Pairwise comparisons using t-tests were used to evaluate any significant overall chi-square or $Q$ tests involving multiple groups. Alpha levels for all comparisons were set at the 0.05 level.

Measures involving continuous data (decision times) or ratings were analyzed using appropriate ANOVA models. Scenario (Military, Firefighter, and Surveillance), Decision-based groupings (Redirectors and Non-redirectors), and Population (ND, USAFA) were considered between-subject factors, while Missions (Peer, Commander, Family Member) or ratings categories/scales from the same instrument/test were treated as within-subject factors in these ANOVAs, unless otherwise noted. All main effects and interactions were followed up with post-hoc paired comparisons evaluated using the Fisher's Least Significant Difference (LSD) test. Fisher's LSD test was employed as it is more powerful than the Bonferroni procedure, despite not controlling as well for family-wise error rates [45]. A less conservative test was deemed most appropriate for this exploratory study since it employed novel Trolley Problem scenarios. Moreover, the practical implications of this study made a less conservative test more appropriate so as to detect true differences, which is a more important goal in this type of study than rejecting false differences [46]. All ANOVA related tests and follow-ups used an alpha level of 0.05.

Effect sizes were calculated for all ANOVAs as partial eta squared (η), a measure reflecting the degree of association between the independent and dependent variables. Partial eta squared values between .01 and .06 are considered small effects, between .06 and .14 medium effects, and above .14, large effects [38].

Table 2. The baseline measures used with both populations.

| Measures | Civilians | Cadets | Populations |
|---|:---:|:---:|:---:|
| **Aggression State Scale** | No Differences | No Differences | No Differences |
| **Triune Ethics (TET) Scale** | | | |
| Wallflower | No Differences | No Differences | No Differences |
| Bunker | No Differences | No Differences | **USAFA > ND** |
| Engagement | No Differences | **Firefighter > Military** | **ND > USAFA** |
| Imaginative | **Surveillance > Military** | No Differences | No Differences |
| **Stress Reactivity Scale** | No Differences | No Differences | **ND > USAFA** |
| Prolonged Reactivity | No Differences | No Differences | No Differences |
| Reactivity to Work Overload | No Differences | No Differences | **ND > USAFA** |
| Reactivity to Social Conflict | No Differences | No Differences | **ND > USAFA** |
| Reactivity to Failure | No Differences | No Differences | **ND > USAFA** |
| Prolonged Reactivity | No Differences | No Differences | **ND > USAFA** |
| **War and Peace Attitudes Scale** | | | |
| Peace Attitudes | No Differences | N/A | N/A |
| War Attitudes | No Differences | N/A | N/A |
| **Altruism** | No Differences | N/A | N/A |
| **Dutifulness** | No Differences | N/A | N/A |
| **Video Game Experience** | | | |
| Age of first video game | No Differences | No Differences | No Differences |
| Hrs. of video games/week | No Differences | No Differences | No Differences |
| Hrs. of video games overall | No Differences | No Differences | **USAFA > ND** |
| Experience with 3D video game | No Differences | No Differences | **USAFA > ND** |

Main measures/scales are in Bold. Subscales are below. N/A means scale was not used with Cadets.

## Baseline measures

Table 2 shows the baseline measures used for both populations and summarizes the results described below. One-way ANOVAs were used to determine if any of the three groups (Military, Firefighter, and Surveillance) differed on any of the overall or subscale scores in either population for the baseline measures shown in Table 2. These comparisons served to verify that the different groups involved in this study were comparable before exposure to the simulations on key attitudes and dispositions that were potentially relevant to simulation performance and reactivity. No significant differences were found across most of the baseline measures administered, except as noted below. For the Reflective Imaginative subscale of the Triune Ethics Meta-Theory (TEM) measure a difference was obtained between the Military and Surveillance group ($p = 0.04$) within the civilian sample. Additionally, differences were obtained between the cadet sample's Firefighter and Military participants within the Engagement subscale of the TEM ($p = 0.05$).

## Main outcome measures

**Proportion of redirect decisions across missions by group and population.** Fig 2 shows the proportion of redirect decisions for civilian (left panel) and cadet populations (right panel) across all three missions (Peer, Commander, Family Member) for the two dilemmic Scenario groups (Military and Firefighter) and Surveillance as control. As the left panel shows, both Military and Firefighter groups in this population showed a drop-off of redirect decisions from Peer to Commander to Family Member missions. In contrast, redirect decisions remained

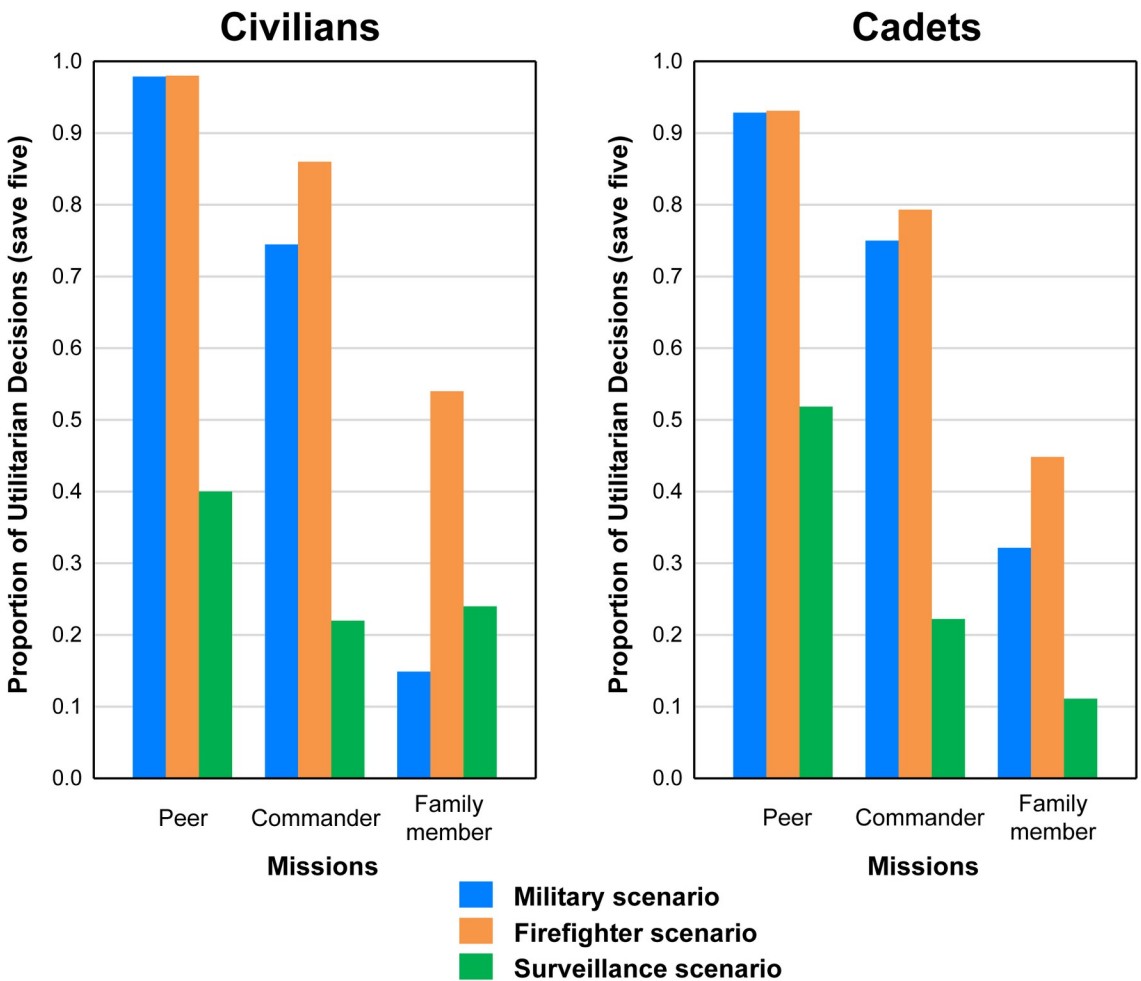

**Fig 2. Decision patterns.** Proportion of utilitarian decision in for Civilians (left side) and Cadets (right side).

consistently low across missions for the Surveillance group. However, for Firefighter group in this population, the drop off in redirect decisions for Family Member mission was much less than was the case for the Military group. For Cadets, the right panel of Fig 2 shows a steady decrease in the proportion of redirect decisions groups across missions that was roughly equivalent for both Military and Firefighter groups. Again, redirect decisions of Cadets in the Surveillance group were relatively low across all missions.

The proportions of utilitarian, redirect decisions made by Civilians on both sides of Fig 2 were evaluated statistically in two ways: Between group tests comparing the two groups (Military, Firefighter and Surveillance) within each mission (Peer, Commander, Family Member) were performed using chi-square tests, with follow up t-tests; within group comparisons for Military and Firefighter group across missions were performed using Cochran's Q tests.

First, for the civilian data on the left side of Fig 1, the overall three group test was significant for both the Commander, $X^2$ (2, $N = 144$) = 48.47, $p < 0.01$, and Family Member missions, $X^2$ (2, $N = 144$) = 19.11, $p < 0.01$. Follow up tests within the Commander mission showed that the difference between Military and Firefighter groups was only marginally significant ($p < 0.10$), but that both Military and Firefighter groups differed significantly from the Surveillance group, $t(114) = 6.07$, $p < 0.01$, and $t(114) = 8.37$, $p < 0.01$, respectively. Follow ups within the

Family Member mission indicated that the difference between the Military and Firefighter groups was significant, t(144) = 4.47, p<0.01, as was the difference between Firefighter and Surveillance groups, $t(114) = 6.07$, $p < 0.01$, but not the difference between Military and Surveillance groups.

Follow-up Cochran's Q tests within the Military and Firefighter groups showed that there was a significant decrease in redirect decisions across the three missions in both scenarios ($Q_M(2) = 60.65$, $p_M < 0.01$; $Q_F(2) = 35.27$, $p_F < 0.01$). These analyses were followed up with paired t-tests. For the Military scenario, the Peer mission differed significantly from both the Commander mission ($t(92) = 3.46$, $p < 0.001$) and Family Member mission ($t(92) = 14.65$, $p < 0.001$). Within group comparisons for the Firefighter scenario produced the same significant differences between Peer and Commander missions, ($t_{PvC}(98) = 2.24$, $p < 0.05$, as well as between Peer and Family Member missions, $t_{PvFM}(98) = 5.95$, $p < 0.001$). In contrast, the Surveillance group showed no significant differences in redirect decisions across the three simulation missions.

Similar comparisons were made between and within groups for the cadet data shown in the right panel of Fig 2. Considering the between group comparisons for Commander and Family Member missions, significant outcomes occurred only for the former mission (X2 (2, N = 74) = 16.97, p < 0.001). Follow up t-tests conducted for the Commander mission showed that there was no significant difference between Military and Firefighter participants. However, both of the dilemmic groups differed significantly from Surveillance: Military (tMvS(45) = -3.77, p < 0.01) and Firefighter (tFvS(46) = -4.32, p < 0.001). This same pattern of significant tests was found within the Peers mission as well, X2 (2, N = 74) = 12.95, p < 0.05. Both Military and Firefighter groups significantly differed from Surveillance (tMvS (45) = -3.04, p < 0.01; tFvS (46) = -3.12, p < 0.01), but not from one another.

For the Military Scenario group, significant decreases in proportions were found across the three missions, Q(2) = 19.14, p < 0.001. Follow up analyses revealed that the Family Member mission was significantly different than the Peers mission (tPvFM(50) = 5.77, p < 0.001) and the Commander mission (tPVc(50) = 3.30, p < 0.01) with a lower proportion of redirect decisions within the Family Member mission; the Peers mission did not differ from the Commander mission within the Military Scenario.

Significant differences across missions were also found for the Firefighter scenario group, Q(2) = 16.42, p < 0.001. This scenario group showed a similar pattern of differences with Peers and Commander mission proportions not differing significantly, but both being significantly different from Family Member (tPvFM(52) = 4.75, p < 0.001; tCvFM(52) = 2.93, p < 0.01).

Finally, significant differences were also found for the Surveillance group, Q(2) = 7.88, p < 0.05. This scenario group showed no differences between Commander and Family Member missions, but both of those missions for this group were significantly different from the Peers mission (tPvC(40) = 2.28, p = 0.03; tPvFM(40) = 3.16, p < 0.01).

The above results confirmed hypothesis (a): The outcome of the Peer mission was consistent with both the traditional paper-and-pencil Trolley Problem applications [1, 3, 9–11] and its more modern applications [22, 23]. As expected, the Surveillance group was roughly a 50/50 split (40/60, to be exact in the civilian sample) for redirect decisions in the baseline TP mission and decreased further, which is in line with the narrative of this scenario, namely to count visitors in a national park. Thus, the Surveillance scenario served well as a non-dilemma, control condition (see Data Availability Statement for data).

Also consistent with hypothesis (a), for both dilemma scenarios, there was a significant decrease in redirect, utilitarian, decisions across missions; showing that, as the value of the one increased, the likelihood of a utilitarian choice decreased. Additionally, the context had a strong impact on the magnitude of the decrease in redirect decisions: participants were much

more likely to make the utilitarian choice, to save five, in the Firefighter scenario than they were in the Military scenario across all three missions. Further, the difference between proportions in the Commander and Family Member mission was much greater within the Military scenario when compared with the Firefighter scenario. The differences using the various tests were significant ($p < 0.01$) both for Civilians and Cadets. No differences were found in these results due to population.

**Decision times across missions by group and population.** As was mentioned previously, only redirect decision times were examined here as participants choosing not to redirect in any mission were assigned the maximum time for that mission. By definition, then, non-redirectors would not be expected to show any variation in decision times across missions.

Fig 3 depicts the mean decision times by population (left panel—Civilians; right panel—Cadets) for redirectors in each dilemma group (Military and Firefighter) for each Mission

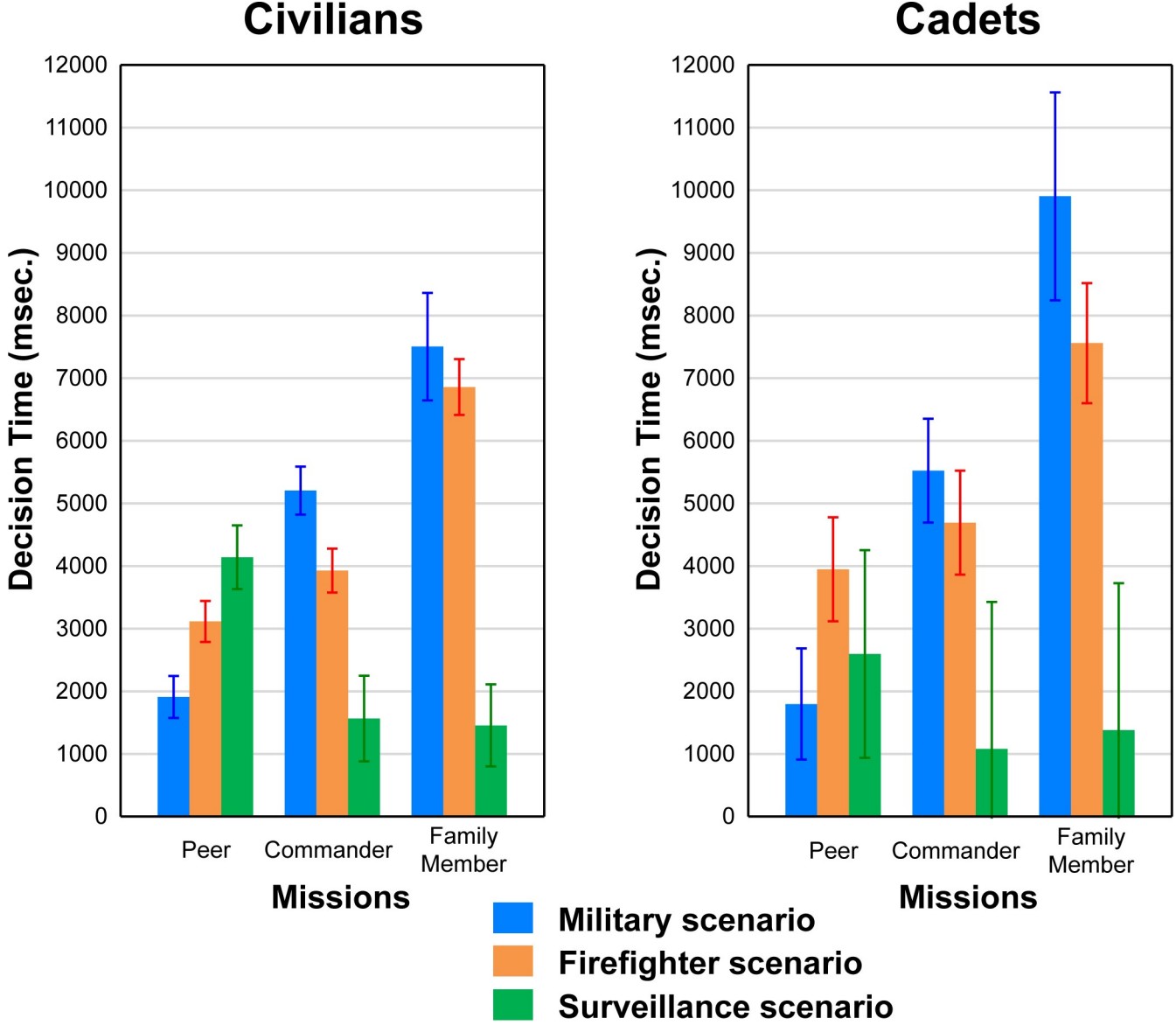

**Fig 3. Decision time results.** Left side: Civilian decision time chart. Right side: Cadet decision time chart.

(Peer, Commander, Family Member). This figure shows two important outcomes for both populations. First, decision times increased notably across missions for Military and Firefighter groups, but not for Surveillance. In terms of these increases for Military and Firefighter, it appears that the decision time for Military increased greatly from Peers to Commander then again from Commander to Family Member. In contrast, Firefighter did not appear to increase notably from Peers to Commander but did from Commander to Family Member. Second, the three groups within each mission all appeared to differ from one another, except for the Family Member mission where Military and Firefighter groups were comparable and both differed from Surveillance. That is, Military and Firefighter groups made quicker decisions than Surveillance within the Peers mission but took longer than Surveillance for both Commander and Family Member missions, in part due to the steady increases for Military and Firefighter groups across missions and in part due to a decrease for Surveillance from Peers to Commander missions.

To examine the pattern of effects in the left panel of Fig 3 statistically, a 3 (Mission) × 3 (Group) ANOVA was applied to the DT data. However, for the reasons noted above regarding changes in the composition of redirector and/or non-redirector groupings across missions, the present analysis treated the Mission variable as a between-subjects factor since the same participants were not redirectors for each mission represented in Fig 2. Therefore, treating Missions as a between-subjects factor in this case was a more conservative choice as between-subject factors typically have more variability than do within-subjects factors.

In this analysis, there were significant main effects of both Scenario and Mission ($F_{(2, 241)} = 12.37$, $p < 0.001$, $\eta2p = 0.09$; $F_{(2, 241)} = 17.31$, $p < 0.001$, $\eta2p = 0.13$; respectively), as well as the interaction effect between Mission and Scenario ($F_{(4, 241)} = 19.49$, $p < 0.001$, $\eta2p = 0.24$). LSD follow-up tests within and across missions revealed that all three Scenario groups (Military, Firefighter, and Surveillance) were significantly different from one another (all $ps < 0.01$) within each mission, except for FM where Military and Firefighter groups were not different. In addition, follow-up tests showed that the Military group increased significantly from Peers to Commander and again from Commander to Family Member missions (all $ps < 0.01$), but Firefighter differed across missions only from Commander to Family Member ($p < 0.001$). Surveillance decreased significantly only from Peers to Commander missions ($p = 0.002$).

As a check that our treatment of Missions as a between-subjects factor in the above analysis did not distort the results, a comparable analysis was run only for unanimous redirectors (URs) in each group. While this reduced the sample size, it permitted Missions to be used as a within-subject factor since the same subject redirected for each mission. The results of this analysis were identical to that reported above.

A comparable ANOVA was applied to the DT data in the right panel. Main effects of Scenario, ($F_{(2, 34)} = 4.41$, $p = 0.019$, $\eta2p = 0.21$), and Missions, ($F_{(2, 34)} = 4.11$, $p = 0.025$, $\eta2p = 0.19$) emerged significant in this analysis, along with a marginally significant Scenario × Missions interaction, ($F_{(4, 34)} = 2.44$, $p = 0.065$, $\eta2p = 0.22$). LSD follow-up tests for the main effect of Scenario indicated that average DT for Military ($p = 0.05$) and Firefighter groups ($p = 0.01$) differed from Surveillance, but did not differ from one another. Likewise, follow-ups to for the main effect of Mission indicated that the average DT for all three missions differed significantly from each other (all $ps < 0.05$). Additional follow ups indicated that the marginally significant interaction occurred because DTs changed significantly from Peers to Family Member missions for both Military and Firefighter groups (both $ps < 0.02$), but not for Surveillance. Moreover, the difference between Military and Firefighter was marginally significant for the Peers mission ($p = 0.08$), but was not significant for either of the other two missions.

The above DT findings are generally consistent with hypothesis (b). DT data showed that the more valuable the one victim became across the three missions (i.e., the more personal the

dilemma became), the longer it took to make the redirect decision, pointing to higher personal and emotional conflict. Results are consistent with previous work showing that dilemma scenarios (Military and Firefighter) provoke emotion [1, 25]. There were no significant differences between the two samples.

**Reasons and justifications for mission decisions.** We examined the reasons and justifications provided by participants in each population for their mission decisions. Reasons were based on the Reasons measure described above and were grouped into three main categories: Philosophical, victim upgrade, or victim downgrade reasons (see Materials section for more detail). Justifications were based on categorizing the post-experiment written statements about their decisions provided by participants as either Philosophical or Non-Philosophical (See Materials section).

Fig 4 shows both reasons (left panel) and justifications (right panel) for both populations in the upper (Civilian) and lower (Cadet) portions of each panel for each group. Only the two dilemmic conditions (Military and Firefighter) are shown here since these groups were the most comparable in terms having to make a significant trade-off decisions involving the sacrifice of lives. In this figure, participants were grouped into "majority redirector" and "majority non-redirector" groups using the two-out-of-three mission decisions criterion described earlier. The upper left panel shows that Civilian majority redirectors endorsed philosophical reasons for hypothetical redirection decisions to a much greater extent than victim upgrade or downgrade reasons. Moreover, Military and Firefighter groups were not much different for any of the reasons categories. The lower left panel indicates that majority redirectors within both Military and Firefighter groups mainly endorsed philosophical reasons for redirect decisions. In contrast, the bottom left panel shows that both Military and Firefighter majority non-redirectors elected higher overall endorsements of arguments in the Upgraded victim category. As would be expected, the Surveillance group redirectors and non-redirectors (not shown) did not exhibit a tendency for elevated reasons in any category.

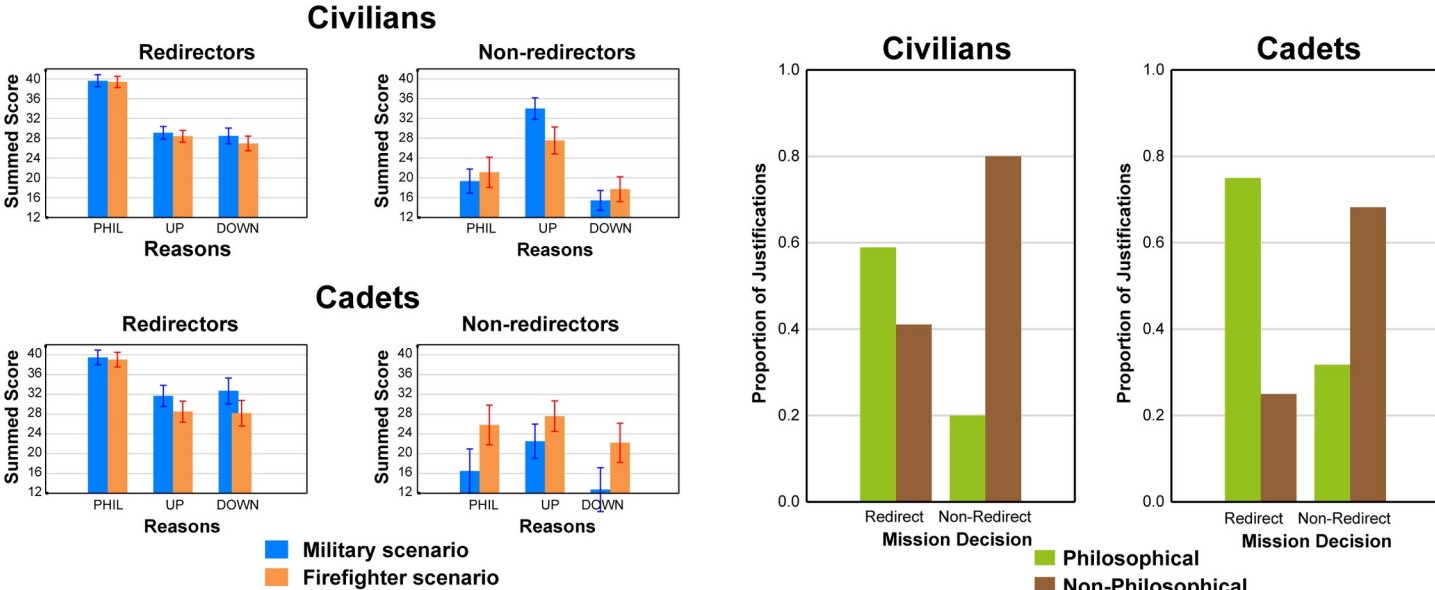

**Fig 4. Reasoning results.** Left side: Summed scores across the four items within each reason type for hypothetical redirection reasons endorsed by majority redirectors (left panel) and for hypothetical non-redirection endorsed by majority non-redirectors (right panel) in two Scenario groups (Military & Firefighter) for both Sample 1 (top panels) and Sample 2 (bottom panels). Right side: Proportion of philosophical justifications for overall Redirectors and Non-Redirectors based on Mission Decision. Justifications were based on the participant's choices made during the simulation for Sample 1 (left panel) and Sample 2 (right panel).

Statistical confirmation for the pattern of effects shown in the upper left (Civilian) portion of Fig 4 was obtained by applying separate 3 × 3 ANOVAs (Reasons: Philosophical, Upgrade, Downgrade vs. Group: Military, Firefighter, and Surveillance) to the reasons for each reason category for both majority redirectors and non-redirectors. Group was a between-subjects factor in these analyses while Reasons was within-subjects. For redirectors, the main effect of reason category was significant, $F_{(2, 166)} = 32.81$, $p < 0.001$, $\eta2p = 0.28$, as was the Reasons × Group interaction, $F_{(4, 166)} = 2.39$, $p = 0.05$, $\eta2p = 0.05$. The main effect occurred because the average score for Philosophical reasons was significantly higher than the average scores for both Upgrade and Downgrade reasons (both ps<0.001), but the latter two means did not differ from one another, as shown by a follow-up LSD test. LSD follow-up tests also showed that the Military and Firefighter groups did not differ from one another.

The lower left portion of Fig 4 shows that majority non-redirectors in both Military and Firefighter groups tallied higher scores in the Upgrade reason category than in the other two categories. In that category, the Military group was higher than the Firefighter group in terms of endorsing upgrade reasons for hypothetical non-redirect decisions. Within the other two reasons categories, Military and Firefighter groups did not appear to differ.

A 3 × 3 ANOVA (Reasons: Philosophical, Upgrade, Downgrade vs. Group: Military, Firefighter, and Surveillance) was applied to the majority non-redirect reasons scores in the lower left panel of Fig 4. Again, Scenario was a between-subjects factor in this analysis while Reasons was within-subjects. The main effect of reason category also was significant in this analysis, $F_{(2, 110)} = 33.90$, $p < 0.001$, $\eta2p = 0.38$, as was the Reasons × Scenario interaction, $F_{(4, 110)} = 3.51$, $p = 0.009$, $\eta2p = 0.11$. The main effect occurred because the average summed score for Upgrade reasons was significantly higher than the average scores for both Philosophical and Downgrade reasons (both ps < 0.001), and the average score for Downgrade reasons was significantly lower than the other two means (both ps < 0.001), as shown by follow-up LSD tests. Follow-up tests also showed that Firefighter differed from Military (p < 0.001) only for Upgrade reason category.

The right panel of Fig 4 also depicts the proportion of Philosophical and non-Philosophical Justifications offered by redirectors and non-redirectors across all three missions for both Civilians (left side) and Cadets (right side). Since no group differences occurred for this measure, Military and Firefighter groups were combined within each population in this figure. From this figure, it appears that for both populations redirect decisions were prompted more philosophical justifications than non-philosophical reasons, while the opposite was true for non-redirect decisions.

To confirm these results statistically, chi-squares was used to compare the proportion of philosophical justifications for redirect decisions across all missions with that of non-redirect decisions in both populations. For Civilians, this analysis yielded a significant outcome, $X2$ (1, N = 252) = 40.32, p < 0.001. Individual t-tests revealed that redirect decisions were more often justified with philosophical than non-philosophical reasons, (t(250) = 6.90, p < 0.001). Conversely, significantly more non-redirect decisions were justified through non-philosophical reasons (p < 0.001). An identical pattern of results were found for the Cadet population. However, no population differences were significant.

For our reasons and justification measures, we compared "redirectors" with "non-redirectors" and found that they differed on the nature of the justifications given for the decisions made. The "redirectors" preferred philosophical reasons such as those involving the greater good, while the "non-redirectors" preferred reasons that referred to personal characteristics of the victim. This applied both to reasons in hypothetical decisions as well as to the justifications of actual decisions made within the simulation, outcomes that are both consistent with hypothesis c) above as well as with previous studies of utilitarian decision-makers [1, 4, 8, 10, 11, 37].

## Baseline measures revisited

It was noted above that several baseline measures were included to check for group difference because of their potential relevance to understanding dilemmic decision-making in situations like those used in the present study. We have already outlined that, with one exception (TET Imagination), there were no baseline group differences in these measures. Here, we report on how participants have been affected by exposure to the simulation experience and how these measures related to decision pattern after the fact. Specifically, the aggression state measure was administered twice, once in baseline and again after the simulation exposure. Also, the Altruism, Dutifulness, War and Peace, and TET baseline measures were included as they might be predictive of whether or not participants would make redirect decisions. To do this, we compared how these baseline measures differed for either the two extreme decision pattern groups (unanimous redirectors, URs, and non-redirectors, UNRs) or the majority decision pattern groups (majority redirectors, MRs, and majority non-redirectors, MNRs; see Materials and Methods, paragraph "Reasoning Measure" for definitions of the two decision pattern groups); groups used for each measure are noted throughout.

**Aggression pre-post.** Pre-post aggression change scores are shown in Fig 5 for the MRs and MNRs in each Scenario group (Military, Firefighter, and Surveillance) for both Civilian (left panel) and Cadet (right panel) populations. Higher numbers indicate increased aggression after simulation exposure. The Majority grouping factor was used here, rather than the Unanimous grouping factor, so as to include all participants. The left panel of this figure shows that for each Civilian Scenario group, redirectors showed a larger post-simulation increase in aggression than non-redirectors. However, Scenario type did not seem to make a difference within either grouping (MRs or MNRs). In contrast, in the right panel (Cadets) it appears that, for both MRs and MNRs, Firefighter participants had the highest change scores, however there does not appear to be much differences across Majority Decision-types.

These effects in the left panel were examined using a 3 (Scenario) $\times$ 2 (Majority grouping) ANOVA. The only significant effect to emerge from this analysis was for the main effect of Majority Decision-maker Type, $F(1, 138) = 5.62$, $p = 0.02$, $\eta^2_p = 0.04$. This effect reflected the fact that regardless of Scenario group, MRs exhibited a greater change in aggression score ($M = 2.05$) than did MNRs ($M = -0.22$). Effect in the right panel were examined through a 3 (Scenario) $\times$ 2 (Majority grouping) ANOVA. No significant nor marginally significant findings were obtained for Cadets.

These results indicate that civilian participants who made a majority of redirect choices had an increase in the state trait of aggression because of their exposure to the simulation relative to non-redirectors. This pattern could relate to what was observed by Greene [3] in his observation of the effects of personal versus impersonal moral dilemmas, but this possibility is diminished because increased aggression applied to all Scenario groups including Surveillance. An alternative possibility here may be related to the active choice. At least for Military and Firefighter groups, redirectors were the only participants making an active choice, sacrificing one to save five. We know from other work [22] that active choices in dilemma conditions generate more emotional responses than do passive choices. However, this would not explain the elevated change for the Surveillance group. Interestingly, however, for Cadets there was no difference between redirectors and non-redirectors on this measure.

**Triune ethics theory.** The Triune Ethics Measure was used to assess each participant's alignment with four different moral orientations: Wallflower, Bunker, Engagement, and Imagination. To see if moral orientation was a useful tool in predicting a participant's willingness to make redirect or non-redirect decisions, separate 3 (Scenario) $\times$ 2 (Unanimous Decision-maker) ANOVAs were run for each moral orientation framework for each population. The

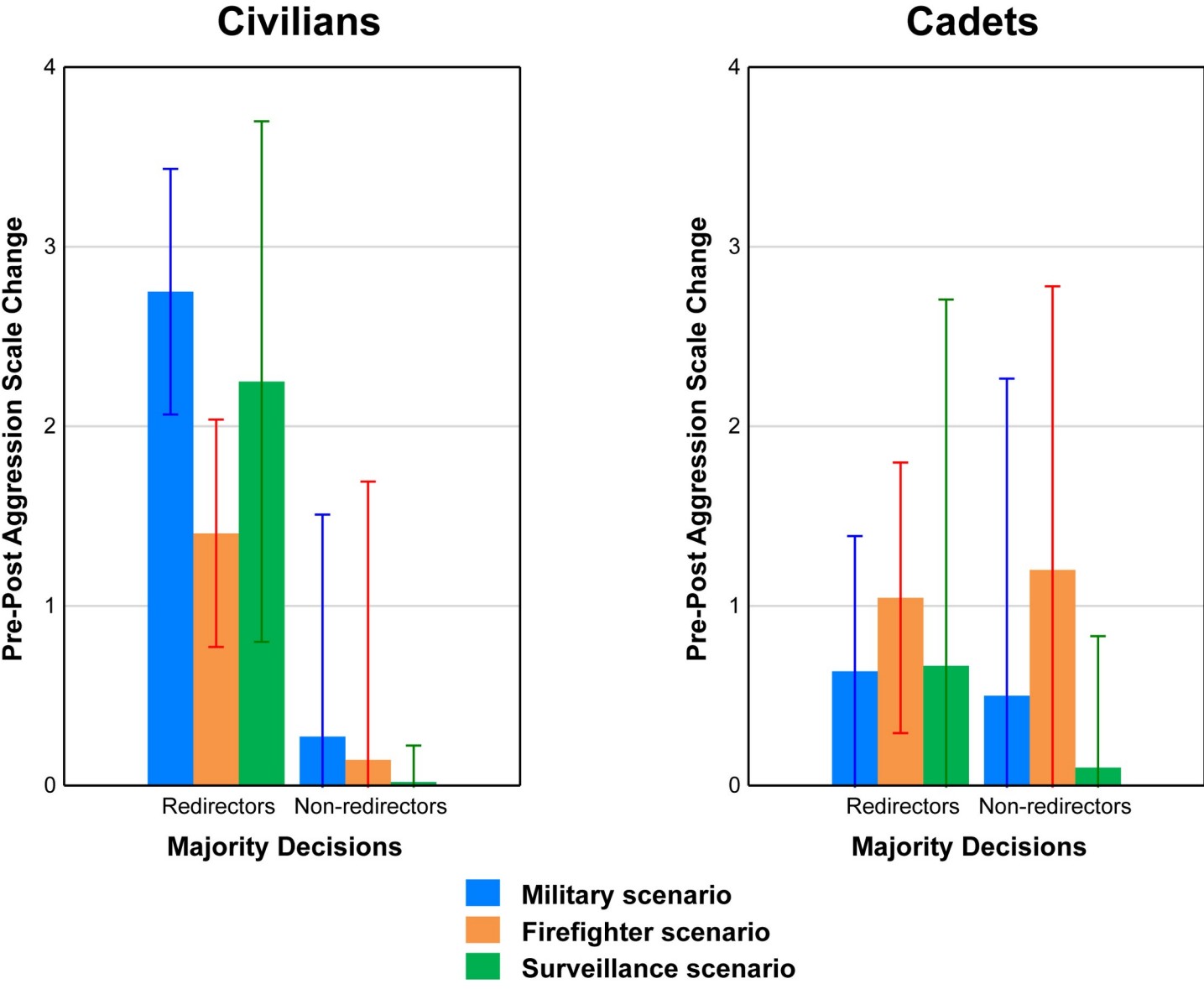

**Fig 5. Aggression state measure.** Pre-post difference in Aggression State score across all three Scenarios within each Mission for Civilians (left panel) and Cadets (right panel).

Unanimous grouping was used here as it was believed that participants who made the same choices across the whole scenario would be much more indicative of a "pure" utilitarian-type or other decision-maker type. For Civilians, no significant effects emerged from any analysis, a finding that suggests that moral orientation did not play a role for Civilians in guiding decisions in this study, at least for the unanimous decision groups (UR and UNR).

For Cadets, however, a significant a main effect for Scenario ($F(2, 68) = 3.282$, $p < 0.05$, $\eta2p = 0.09$) was found only for the Engagement orientation (Caring, Compassionate, Merciful, and Cooperative) such that Firefighter participants aligned more with this orientation than Military ($p = 0.05$). A significant main effect of majority decision-maker type also was seen for the Imaginative orientation (Reflective, Thoughtful, Inventive, and Reasonable; $F(1, 68) = 7.25$, $p < 0.01$, $\eta2p = 0.10$) such that Majority Non-Redirectors had greater alignment with this

orientation than did Majority Redirectors (p < 0.01). These outcomes for Cadets could have something to do with the military training of cadets that might sensitize them differently to military vs. firefighting situations.

**Altruism.** Altruism, or willingness to help, was another measure investigated to determine its predictive value for unanimous decision-making patterns. It is possible that participants higher in Altruism would be more likely to make utilitarian, altruistic, decisions based on the concept of the greater good, especially in the P Mission. The trait of altruism is one often observed in participants who make the utilitarian choice across various Trolley Problem tasks [4, 8, 34, 47], which is why Unanimous decision types were used here. However, a 3 (Scenario) × 2 (Unanimous Decision-maker) ANOVA of this measure failed to reveal any significant effects. Thus, for Civilians, altruism was not predictive of decision pattern, at least for unanimous decision makers. This scale was not administered to Cadets because of time constraints.

**Dutifulness.** Dutifulness, or willingness to obey orders, was also included as a measure because of its potential to predict utilitarian decision-making (i.e., redirecting in this study) which is why Unanimous decision type was used here. Based on previous studies [4, 8], it is possible that higher levels of dutifulness would predict a greater likelihood that participants would make the greater-good decision of redirecting to save the five. To assess this, a 3 (Scenario) × 2 (Unanimous Decision-maker) ANOVA was conducted. Again, no significant findings were obtained for Civilians suggesting that willingness to obey orders was not related to decision making in this study. Again, this scale was not administered to Cadets.

**War and peace attitude.** A measure looking at war and peace attitudes was also utilized in this study with Civilians but not for Cadets. This scale produced two separate scores for each attitude. It was possible that attitudes on this scale could influence the decisions to redirect or not within our simulation. For instance, if a participant were to have more positive attitudes towards peace with significantly lower levels in their attitude towards war, they might display more passivist actions, deciding not to redirect in order to not have to take responsibility for the death of another as framed by the ME HURT YOU heuristic [24, 34, 49].

To test their predictive values for Civilians, two separate 3 (Scenario) × 2 (Unanimous Decision-maker) ANOVAs were run for War attitudes and Peace attitudes. No significant findings were observed for War attitudes. However, there was a significant interaction effect seen for Peace Attitudes between Scenario and Unanimous Decision-maker type, $F(2, 48) = 4.60$, $p = 0.01$, $\eta^2_p = 0.16$. Follow up tests showed that this interaction occurred because Peace attitude scores were significantly higher ($p = 0.01$) for the UNRs than for URs, but only within the Firefighter scenario group.

Overall, it also appeared that Civilian non-redirectors displayed higher Peace attitude scores than their redirector counterparts within the Firefighter scenario group. The above explanation of the effect of Peace attitudes might make sense here as those who did not redirect, the deontologists, had more positive attitudes toward Peace ideals, which aligns with passivity. This passive tendency may correlate with the fact that the non-redirect choice in our simulation was an inactive choice. Furthermore, it aligns with the basic moral rule that it is wrong to kill and to make the active choice of redirecting the missile would be straying from that rule. It is interesting, however, that this pattern only occurred within the Firefighter scenario. Perhaps this was the case since this scenario is less associated with the dichotomy of war and peace, while the Military scenario forces the participant to confront this dichotomy since war-time characters are present.

## Discussion

The decision outcomes we obtained with the Peers missions in this study under both Military and Firefighter scenarios corroborated the basic Trolley Problem finding that a majority of

decision-makers chose to redirect towards the one to save the five. This outcome validates the dilemmic missions in our simulation as being reliable variations on the classic Trolley Problem dilemma [1, 3, 10, 11, 22, 23, 25, 35, 48]. Moreover, in the non-dilemmic Surveillance scenario, we observed close to a fifty-fifty split suggesting that this aspect of the simulation served as an effective control condition. Finally, across both populations in this study, similar decision results were found as evidenced by very few significant differences between Civilians and Cadets. These finding were generally consistent with our first hypothesis.

In this study, we also increased the social and emotional value of the one across three consecutive dilemmic scenarios. Our hypothesis here was that we would replicate findings reported throughout moral decision-making literature with personal versus impersonal moral dilemmas such as the Trolley versus Footbridge Problem [9, 25, 35, 49] showing that, as moral dilemmas became more personal in nature, the number of redirect decisions decrease. This outcome was indeed observed in the present study as evidenced by significant drops in proportions of redirect decisions across our missions as the one's social and emotional value increased across the Commander and Family Member missions. This change across mission was not obtained in the control condition, however.

The use of a "commander" to increase the value of the one is a unique feature of this study. There is, of course, existing literature which increases the value of the one, as this study does, but not in the same personal and professional way as by using a commander or fire chief [4, 9, 12]. The closest one can find in contemporary literature is that of a universal figurehead, such as a world leader, but not a "personal" figurehead, such as a boss or superior. This use of a "personal" figurehead introduces a potentially new kind of dilemma in need of further exploration. For instance, if the one was the president or prime minister of the decision-maker's country, would such a "global" commander be a more obvious candidate to be saved? The value we attach to global (e.g., Presidents) vs. personal (e.g., bosses) leaders needs to be better understood. In terms of our original expectations, it was not clear where a "commander" would fit in. Would a commander be more like a peer or more like a family member? The present findings suggest the former rather than the latter.

Civilians and military personnel seem to have perceived commanders differently in this study, however. Results for the Commander mission within the civilian sample suggest that military commander's role was viewed as more important than that of the Firefighter commander, as evidenced by the fact that participants saved the Military commander more often than the Firefighter commander. Further, in their reasoning they cited the importance of the Military commander's role in mission success. However, USAFA cadets made no such distinction between commanders, valuing the roles of both types of commanders (military or firefighters) equally.

Recent research has shown that people generally perceive soldiers differently than regular Civilians in terms of character traits [50, 51]. The present study extends this research in showing that Civilians also may perceive military and firefighter personnel differently, possibly taking a more pragmatic approach to soldiers than to firefighters. Evidence for this comes from the Firefighter scenario, where participants (especially Civilians) more often elected to save five firefighters over a family member. As one participant reported, "firefighters have families at home." This type of argument was not reported for saving the five soldiers in the Military scenario, instead appealing to utilitarian reasons when a redirect decision was made in this condition. Furthermore, as was evidenced in the Military scenario, Civilians viewed sacrifice as the soldier's duty, an idea echoed in recent surveys [52]. Thus, Civilians may feel more emotional engagement with dilemmas involving firefighters over military personnel [52, 53]. This personalization shift may be a byproduct of a post 9–11 United States.

Based on other work [3, 34], we also hypothesized that decision time (DT) measures would be affected in a specific ways by the dilemmas embedded in our simulations. Specifically, we

hypothesized that more personal dilemmas would result in longer DTs, because they increase decision conflict. Accordingly, we expected that redirect decisions in Family Member mission would take longer than similar decisions in the Peers mission, with the Commander mission falling in between. These increases across mission were expected to occur mainly within our two dilemmic conditions, not in the control condition. These expectations were confirmed for both populations in the present study with the finding that DTs for Military and Firefighter redirect decisions were significantly longer for the Family Member mission when compared to the Peers mission. This pattern across missions was not evident for the surveillance groups in either population. Redirect DTs in the Commander mission indeed were in between those of the Peers and Family Member missions for both Military and Firefighter scenarios.

Our results indicate that redirectors provided different justifications than non-redirectors, as we expected. Overall, redirectors made more philosophical justifications than non-redirectors. Interesting differences, both quantitative and qualitative, were found when comparing Military and Firefighter scenario results across both populations' simulation decision justifications. These differences were most apparent within the Commander and Family Member missions when observing differences in reasons given for the decisions made.

## Limitations

The present study has several limitations that should be noted. Broadly, these limitations fall into the categories of study design and data collection/management. Regarding design, the order of the three missions within the Simulation was fixed. The Peer mission was always first, followed by the Commander mission, and then the Family Member mission. This led to concerns of order effects within the Simulation. By having the Missions in the same order throughout, it is possible that participants would become desensitized to the Trolley Problem decision by the time the Family Member mission occurred which could have affected outcomes within our measures of emotional conflict (DT) due to issues such as participant fatigue. In fact, some evidence of this was provided by responses to the Mission Review Debriefing.

Additionally, in terms of the missions, a limitation occurred due to the go/no-go nature of responses to simulation decisions; across all three missions and in all scenarios, the active choice was paired with the choice to redirect which prevented us from making many solid conclusions about the nature of these deontological, non-redirect, decisions. For instance, we were unable to record DTs for these decisions as they occurred as a default when the decision timer ran out. Furthermore, as Navarrete et al. [22] concluded in their study, active responses inherently produce different outcomes in terms of emotional activation and investment in decision than to inactive responses. This active-redirect pairing could, therefore, not only affect the measures taken during the simulation for non-redirect decisions, but also could have influenced the reasons survey as well.

A final design issue relates to the fact that the hypothetical reasons measure was administered only at the end, which means these measures could not properly detect any effects that were evoked during specific missions within a scenario. At best, these measures reflected a participant's overall experience with all of the missions to which they were exposed.

Finally, due to experimenter error, software/hardware malfunction, and study runtime constraints, both samples used in this study were affected by data loss. Of the 172 participants run for sample 1 (Civilians), only 144 had complete data. For sample 2 (Cadets), only 75 participants had complete data out of the 98 runs. This sample in particular suffered greatly from runtime constraints due to the strict day-to-day schedules of the USAFA cadets, further, the research team at this location was much smaller overall.

## Conclusions

Despite its limitations, the present study demonstrates that Trolley-Problem-like dilemmas can be embedded in more modern and relevant contexts with comparable outcomes to those obtained with the original paper-and-pencil methods. These findings both confirm and extend the results of previous studies in the area of trolleyology research using a novel RPA-like 3D simulation to which both military-oriented and civilian populations were exposed. Our confirmatory evidence comes from finding the same tendency in both populations to sacrifice one to save five in our novel simulated dilemma tasks, just as is found in response to the traditional paper and pencil Trolley Problem situation. In addition, we have replicated previous findings that increasing the value of the one to be sacrificed changes the willingness to save five at the expense of the one.

Our results indicate surprisingly strong effects of context on dilemmatic decision-making. We have extended previous findings in showing that those who are more invested in a particular context or in its hypothetical victims will react differently and offer different justifications for their actions in the face of ostensibly similar dilemmas. This result means that moral decisions are not always made using a fixed standard, but rather are made using the standards that are implied or embedded in contexts in which those decisions occur. Tragic decisions made in a military context may be different from those made in the face of an apparently similar dilemma in a context based on firefighting. While the present findings underscore the important role context plays in decision-making, more research is needed to empirically uncover continuities and discontinuities between different contexts in which moral decisions occur [53].

## Acknowledgments

The authors would like to thank the University of Notre Dame's eMotion and eCognition and moral psychology laboratories, as well as the Sensation and Perception Lab at the US Air Force Academy.

**Disclaimer:** The views of the authors are their own and do not purport to reflect the position of the United States Air Force Academy, the Department of the Air Force, or the Department of Defense.

## Author Contributions

**Conceptualization:** Markus Christen, Darcia Narvaez, Michael Villano, Charles R. Crowell.

**Data curation:** Julaine D. Zenk, Michael Villano, Daniel R. Moore.

**Formal analysis:** Markus Christen, Julaine D. Zenk, Charles R. Crowell.

**Funding acquisition:** Markus Christen, Charles R. Crowell, Daniel R. Moore.

**Investigation:** Markus Christen, Julaine D. Zenk, Michael Villano, Daniel R. Moore.

**Methodology:** Darcia Narvaez, Michael Villano, Charles R. Crowell.

**Project administration:** Markus Christen.

**Resources:** Charles R. Crowell, Daniel R. Moore.

**Software:** Michael Villano.

**Supervision:** Darcia Narvaez, Charles R. Crowell.

**Validation:** Charles R. Crowell.

**Visualization:** Markus Christen, Michael Villano, Charles R. Crowell.

**Writing – original draft:** Markus Christen.

**Writing – review & editing:** Darcia Narvaez, Julaine D. Zenk, Michael Villano, Charles R. Crowell, Daniel R. Moore.

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
