## [Decision Letter · Decision Letter 0]

5 Nov 2020

PONE-D-20-30443

Trolley Dilemma in the Sky: Context Matters When Civilians and Cadets Make Remotely Piloted Aircraft Decisions

PLOS ONE

Dear Dr. Christen,

Thank you for submitting your manuscript to PLOS ONE. After careful consideration, we feel that it has merit but does not fully meet PLOS ONE’s publication criteria as it currently stands. Therefore, we invite you to submit a revised version of the manuscript that addresses the points raised during the review process.

Please find below the reviewer's comments, as well as those of mine.

We look forward to receiving your revised manuscript.

Kind regards,

Valerio Capraro

Academic Editor

PLOS ONE

Journal Requirements:

Reviewers' comments:

Reviewer's Responses to Questions

**Comments to the Author**

1. Is the manuscript technically sound, and do the data support the conclusions?

Reviewer #1: Partly

2. Has the statistical analysis been performed appropriately and rigorously? 

Reviewer #1: I Don't Know

3. Have the authors made all data underlying the findings in their manuscript fully available?

Reviewer #1: Yes

4. Is the manuscript presented in an intelligible fashion and written in standard English?

Reviewer #1: No

5. Review Comments to the Author

Reviewer #1: I recently had the pleasure of reading the paper “Trolley Dilemma in the Sky: Context Matters When Civilians and Cadets Make Remotely Piloted Aircraft Decisions”, submitted for publication in PLOS ONE.

I think the study presented in this paper is intriguing and has many things going for it, but there are also a number of ways to improve it. Not all of these will be as essential as others (and some may not be possible within the scope of the paper/journal), and I will try to indicate which I think are more important; but ultimately I leave it to the editor to decide whether to invite a revision (and if so, which aspects it will be most important to revise).

My comments are presented below in roughly the order they occurred to me while reading the paper. (Please excuse the terseness; it’s been a long year.)

Always signed,

Hanne M Watkins

***

Abstract:

“We found that participants (Air Force cadets vs. civilian students)...”

This would probably be less confusing if the “vs.” was replaced with “and”.

If there’s room in the word count, indicate that the “the value of the one increased” by going from peer to authority figure to family?

“However, in the rescue context…” relative to which other context(s)?

**Introduction**

Overall comment: somewhat dense and imprecise writing - consider having someone not involved in the project previously read over it for a fresh perspective? For example, “The Trolley Problem has become a cornerstone in decision-making research” - probably only in moral judgment/decision-making research, really?

Overall comment: there are some significant gaps in the literature review (or at least in the papers cited). I’ve made some suggestions below.

Minor comment: There are different versions of the trolley problem, and only the “switch” version has the structure described in the second sentence of the introduction. See e.g., the many versions in Moore, A. B., Clark, B. A., & Kane, M. J. (2008). Who shalt not kill? Individual differences in working memory capacity, executive control, and moral judgment. Psychological science, 19(6), 549-557.

Additional papers to potentially cite on the way that characteristics of potential victims influence judgments/decisions in TP dilemmas:

Petrinovich, L., O'Neill, P., & Jorgensen, M. (1993). An empirical study of moral intuitions: Toward an evolutionary ethics. Journal of personality and social psychology, 64(3), 467.

Petrinovich, L., & O'Neill, P. (1996). Influence of wording and framing effects on moral intuitions. Ethology and Sociobiology, 17(3), 145-171.

And more generally (though less relevantly) how relationships influence moral judgments:

Simpson, A., Laham, S. M., & Fiske, A. P. (2016). Wrongness in different relationships: Relational context effects on moral judgment. The Journal of social psychology, 156(6), 594-609.

Line 44: “No study thus far has systematically analyzed whether the context of the decision problem matters as well” - I don’t think that’s quite accurate. This paper (full disclosure: I wrote it) investigates the context of war (vs peace) using TP dilemmas, making similar arguments to the present paper:

Watkins, H. M., & Laham, S. (2019). The influence of war on moral judgments about harm. European Journal of Social Psychology, 49(3), 447-460.

And in this paper I make a more general argument about the same thing (again, consistent with the present paper’s argument): Watkins, H. M. (2020). The morality of war: A review and research agenda. Perspectives on Psychological Science, 15(2), 231-249.

See also this introduction to a special issue of perspective:

Schein, C. (2020). The importance of context in moral judgments. Perspectives on Psychological Science, 15(2), 207-215.

Another couple of papers critiquing the TP for being unrealistic or uncommon (could be added where the authors already cite some critiques):

Bloom, P. (2011). Family, community, trolley problems, and the crisis in moral psychology. The Yale Review, 99(2), 26-43.

Bauman, C. W., McGraw, A. P., Bartels, D. M., & Warren, C. (2014). Revisiting external validity: Concerns about trolley problems and other sacrificial dilemmas in moral psychology. Social and Personality Psychology Compass, 8(9), 536-554.

Line 59: “These decisions may involve trolley-like moral dilemmas where the RPA operator and crew must choose between (a) killing notorious terrorists and sacrificing nearby innocent bystanders, or (b) doing nothing, which will spare bystanders but enable terrorists to commit future acts of violence.” This is a fine description of the dilemma the RPA operator faces, but it’s not a perfect match for the structure of classic trolley problems. I think a bit more time could be spent on the differences and similarities, especially because this section made me expect that the study in the present paper would also present a dilemma between innocent bystanders and terrorists.

Also, this exact dilemma has been examined in this study:

Malle, B. F., Magar, S. T., & Scheutz, M. (2019). AI in the sky: How people morally evaluate human and machine decisions in a lethal strike dilemma. In Robotics and Well-Being (pp. 111-133). Springer, Cham.

**Materials and Methods**

The experimental design is clear, but I found the “C-mission, F-mission, P-mission” labels confusing. It might be good to find a more transparent way to describe these missions.

The details about the simulation (using Half-Life 2 SDK) on line 67 are the kind of things I would expect to see in the Materials and Methods section, rather than in the introduction. (And instead, in the intro, I would probably expect a bit more of an elaboration of how the hypotheses were derived from the theory.)

Another hypothesis is introduced at line 130. I think it would be good to have all the hypotheses together (and more closely tied to theory/previous research).

Were the hypotheses and data analysis plans pre-registered? This could be considered (for example using a website like AsPredicted.org) for future studies on this topic.

Why did the authors assume the effect sizes were large?

I like the inclusion of a control condition that involves “shooting” (with a camera!) five people vs one person. Here’s a relevant paper: Crone, D. L., & Laham, S. M. (2017). Utilitarian preferences or action preferences? De-confounding action and moral code in sacrificial dilemmas. Personality and Individual Differences, 104, 476-481.

Why is the information about the sample included twice? Overall, the structure of this section is very confusing, and some parts are repetitive.

A lot of measures are listed in this section, and then not discussed in the results section. Suggest moving measures that are not used again to supplemental materials (and referring to them in main manuscript).

** Results **

(I’m sorry, I ran out of time to do this section full justice)

The “statistics sample 1” and “statistics sample 2” paragraphs should go in the results section.

I would find it helpful if the results section included more basic, descriptive, statistics for all the outcome measures (perhaps in a table) before launching into the inferential statistics.

I’m slightly confused by the statistical analysis choices. I think a multilevel analysis might be appropriate since the person involved (peer, commander, family) was a repeated measure (i.e., each participants completed all three), while the type of scenario (military, firefighter, and surveillance) was between-subjects. Some of the comparisons made in the text are not supported by analyses.

For the “redirectors” vs “non-redirectors” comparisons, were the two samples combined?

** Discussion and Conclusion **

Line 444: “...possibly taking a more pragmatic approach to soldiers”

This paper makes a conceptually similar point (without a comparison to fire fighters): Watkins, H. M., & Laham, S. M. (2020). The principle of discrimination: Investigating perceptions of soldiers. Group Processes & Intergroup Relations, 23(1), 3-23.

The organization of this section is confusing as well - limitations are mentioned early on, and then again under a separate subheading “limitations”.

I’d avoid using the word “priming/primed” (line 485, line 66), as it implies something other than what this study actually does.

I think the authors undersell the general interest and importance of their study, as it does add a lot of interesting nuance to classic trolleyology.

6. PLOS authors have the option to publish the peer review history of their article (what does this mean?). If published, this will include your full peer review and any attached files.

Reviewer #1: **Yes: **Hanne M Watkins

---

## [Author Response · Author response to Decision Letter 0]

24 Dec 2020

Trolley Dilemma in the Sky: Context Matters When Civilians and Cadets Make Remotely Piloted Aircraft Decisions – rebuttal letter

In this document, we outline in detail how we have addressed the concerns and critique of the reviewers. All changes in the manuscripts are marked in yellow in the document “Revised Manuscript with Track Changes” (we refrained to mark orthographical corrections for better readability).

We refrained from depositing our laboratory protocols in protocols.io as we consider the paper itself as well as the data provided on zenodo.org (including metadata-explanation) as sufficient to enhance the reproducibility of your results. 

Editor remarks

We have checked the guidelines and made the requested changes.

We note that you have stated that you will provide repository information for your data at acceptance. Should your manuscript be accepted for publication, we will hold it until you provide the relevant accession numbers or DOIs necessary to access your data. If you wish to make changes to your Data Availability statement, please describe these changes in your cover letter and we will update your Data Availability statement to reflect the information you provide.

The data is available at Zenodo (www.zenodo.org); DOI: 10.5281/zenodo.4391312

PLOS authors have the option to publish the peer review history of their article (what does this mean?). If published, this will include your full peer review and any attached files.

We agree with publishing the peer review history if the reviewer does so as well.

We have checked and adapted all figure files using the tool mentioned by the editor. In particular, we checked for size (<10 MB), color mode (RGB), and fonts used (Arial).

Reviewer #1

Abstract: “We found that participants (Air Force cadets vs. civilian students)...” This would probably be less confusing if the “vs.” was replaced with “and”. If there’s room in the word count, indicate that the “the value of the one increased” by going from peer to authority figure to family? (…) “However, in the rescue context…” relative to which other context(s)?

We have corrected the abstract as suggested by the reviewer.

Introduction - Overall comment: somewhat dense and imprecise writing - consider having someone not involved in the project previously read over it for a fresh perspective? For example, “The Trolley Problem has become a cornerstone in decision-making research” - probably only in moral judgment/decision-making research, really?

We would like to thank the reviewer for her very helpful comments and suggestions for additional references. We have rewritten and complemented large parts of the introduction following the advices of the reviewer.

Overall comment: there are some significant gaps in the literature review (or at least in the papers cited). I’ve made some suggestions below.

As mentioned: we are grateful for the recommendations of the reviewer and have implemented most of them in order to close the gaps in the literature review. Some of the gaps resulted from the fact that the manuscript in an earlier version has been submitted to a journal with strict length limitations. 

Minor comment: There are different versions of the trolley problem, and only the “switch” version has the structure described in the second sentence of the introduction. See e.g., the many versions in: 

- Moore, A. B., Clark, B. A., & Kane, M. J. (2008). Who shalt not kill? Individual differences in working memory capacity, executive control, and moral judgment. Psychological science, 19(6), 549-557.

We thank the reviewer for this suggestion and we have quoted and added the reference in the rewritten introduction.

Additional papers to potentially cite on the way that characteristics of potential victims influence judgments/decisions in TP dilemmas:

− Petrinovich, L., O'Neill, P., & Jorgensen, M. (1993). An empirical study of moral intuitions: Toward an evolutionary ethics. Journal of personality and social psychology, 64(3), 467.

− Petrinovich, L., & O'Neill, P. (1996). Influence of wording and framing effects on moral intuitions. Ethology and Sociobiology, 17(3), 145-171.

And more generally (though less relevantly) how relationships influence moral judgments:

− Simpson, A., Laham, S. M., & Fiske, A. P. (2016). Wrongness in different relationships: Relational context effects on moral judgment. The Journal of social psychology, 156(6), 594-609.

We have added remarks and citations referring to those papers in the introduction.

Line 44: “No study thus far has systematically analyzed whether the context of the decision problem matters as well” - I don’t think that’s quite accurate. This paper (full disclosure: I wrote it) investigates the context of war (vs peace) using TP dilemmas, making similar arguments to the present paper:

− Watkins, H. M., & Laham, S. (2019). The influence of war on moral judgments about harm. European Journal of Social Psychology, 49(3), 447-460.

And in this paper I make a more general argument about the same thing (again, consistent with the present paper’s argument): 

− Watkins, H. M. (2020). The morality of war: A review and research agenda. Perspectives on Psychological Science, 15(2), 231-249.

See also this introduction to a special issue of perspective:

− Schein, C. (2020). The importance of context in moral judgments. Perspectives on Psychological Science, 15(2), 207-215.

Another couple of papers critiquing the TP for being unrealistic or uncommon (could be added where the authors already cite some critiques):

− Bloom, P. (2011). Family, community, trolley problems, and the crisis in moral psychology. The Yale Review, 99(2), 26-43.

− Bauman, C. W., McGraw, A. P., Bartels, D. M., & Warren, C. (2014). Revisiting external validity: Concerns about trolley problems and other sacrificial dilemmas in moral psychology. Social and Personality Psychology Compass, 8(9), 536-554.

We have added remarks and citations referring to those papers in the introduction and the conclusion.

Line 59: “These decisions may involve trolley-like moral dilemmas where the RPA operator and crew must choose between (a) killing notorious terrorists and sacrificing nearby innocent bystanders, or (b) doing nothing, which will spare bystanders but enable terrorists to commit future acts of violence.” This is a fine description of the dilemma the RPA operator faces, but it’s not a perfect match for the structure of classic trolley problems. I think a bit more time could be spent on the differences and similarities, especially because this section made me expect that the study in the present paper would also present a dilemma between innocent bystanders and terrorists.

Thank you for this important remark. We have clarified in the rewritten introduction that this type of dilemma, which RPA pilots indeed can experience in their operations, is not equivalent to the classic trolley problem. We also clarified that the dilemmas we used do not match this “terrorist” vs. “bystander” problem, but is actually in line with the classic trolley problem (in the military context, it can be understood as a “friendly fire” dilemma).

Also, this exact dilemma has been examined in this study:

− Malle, B. F., Magar, S. T., & Scheutz, M. (2019). AI in the sky: How people morally evaluate human and machine decisions in a lethal strike dilemma. In Robotics and Well-Being (pp. 111-133). Springer, Cham.

We have added a remark and a citation referring to this paper in the introduction.

Materials and Methods: The experimental design is clear, but I found the “C-mission, F-mission, P-mission” labels confusing. It might be good to find a more transparent way to describe these missions.

We have changed the labels and now use consistently the terms “Peer mission”, “Commander mission” and “Family Member mission”.

The details about the simulation (using Half-Life 2 SDK) on line 67 are the kind of things I would expect to see in the Materials and Methods section, rather than in the introduction. (And instead, in the intro, I would probably expect a bit more of an elaboration of how the hypotheses were derived from the theory.)

We have elaborated more on how the simulation was created in the Materials and Methods section (page 7) and we outlined how the hypotheses were derived from the theory in the introduction on pages 4-5).

Another hypothesis is introduced at line 130. I think it would be good to have all the hypotheses together (and more closely tied to theory/previous research).

Thank you for this suggestion. To provide a better overview of our hypotheses, we have put them into a table. In the text, we linked them to the cited literature.

Were the hypotheses and data analysis plans pre-registered? This could be considered (for example using a website like AsPredicted.org) for future studies on this topic.

Although the hypotheses and data analysis plan were derived at the beginning of the study, we unfortunately did not preregister them, as the conceptualization of the study started in a time when the issue of preregistration just emerged as a major methodological issue. For future studies, we will make use of preregistration.

Why did the authors assume the effect sizes were large?

On page 6 we have clarified our reasoning regarding effect size. We also discussed this aspect in the new “analysis strategy” paragraph in the Results section (pages 10-11).

I like the inclusion of a control condition that involves “shooting” (with a camera!) five people vs one person. Here’s a relevant paper: 

− Crone, D. L., & Laham, S. M. (2017). Utilitarian preferences or action preferences? De-confounding action and moral code in sacrificial dilemmas. Personality and Individual Differences, 104, 476-481.

We thank the reviewer for this suggestion: We have added the reference as well as a short comment on page 7.

Why is the information about the sample included twice? Overall, the structure of this section is very confusing, and some parts are repetitive.

We have optimized the structure and made sure that information on the sample size only appears once.

A lot of measures are listed in this section, and then not discussed in the results section. Suggest moving measures that are not used again to supplemental materials (and referring to them in main manuscript).

We made a substantial revision of the Result section. We decided to report also the results of the other measures and thus kept the information on those measures in the Materials and Methods section.

Results: The “statistics sample 1” and “statistics sample 2” paragraphs should go in the results section.

We have moved those parts to the result section and we also made significant changes in the Result section to clarify various point. 

I would find it helpful if the results section included more basic, descriptive, statistics for all the outcome measures (perhaps in a table) before launching into the inferential statistics.

We thank the reviewer for his suggestion. We used a different approach by largely rewriting and restructuring the Result section to clarify those and other issues.

I’m slightly confused by the statistical analysis choices. I think a multilevel analysis might be appropriate since the person involved (peer, commander, family) was a repeated measure (i.e., each participants completed all three), while the type of scenario (military, firefighter, and surveillance) was between-subjects. Some of the comparisons made in the text are not supported by analyses.

In the beginning of the new Result Section, we added a paragraph for explaining the statistical analysis choices. We also clarified that we did use repeated measures where this was possible.

For the “redirectors” vs “non-redirectors” comparisons, were the two samples combined?

On page 4, we noted that the two samples were not combined for this analysis.

Discussion and Conclusion: Line 444: “...possibly taking a more pragmatic approach to soldiers”. This paper makes a conceptually similar point (without a comparison to fire fighters): 

- Watkins, H. M., & Laham, S. M. (2020). The principle of discrimination: Investigating perceptions of soldiers. Group Processes & Intergroup Relations, 23(1), 3-23.

We added this reference and acknowledged this line of research. We also mentioned that our study extends this research by showing that civilians my view soldiers differently from firefighters.

The organization of this section is confusing as well - limitations are mentioned early on, and then again under a separate subheading “limitations”.

We thank the reviewer for this observation. We have partly rewritten and restructured this section to avoid redundancies.

I’d avoid using the word “priming/primed” (line 485, line 66), as it implies something other than what this study actually does.

We have eliminated the use of “priming/primed” in the paper.

I think the authors undersell the general interest and importance of their study, as it does add a lot of interesting nuance to classic trolleyology.

We thank the reviewer for this evaluation; we have added a concluding section to strengthen the contribution of our study.

---

## [Editor Report · Decision Letter 1]

26 Jan 2021

PONE-D-20-30443R1

Trolley Dilemma in the Sky: Context Matters When Civilians and Cadets Make Remotely Piloted Aircraft Decisions

PLOS ONE

Dear Dr. Christen,

Thank you for submitting your manuscript to PLOS ONE. After careful consideration, we feel that it has merit but does not fully meet PLOS ONE’s publication criteria as it currently stands. Therefore, we invite you to submit a revised version of the manuscript that addresses the points raised during the review process.

We look forward to receiving your revised manuscript.

Kind regards,

Valerio Capraro

Academic Editor

PLOS ONE

Additional Editor Comments (if provided):

Unfortunately, the reviewer who reviewed the first version of the manuscript is unable to review the revision. However, I have read the response letter and I am satisfied with the way you addressed their comments. However, I have noticed that you did not address the comments I gave on the original version of the manuscript, which I paste here: "I have now collected one review from one expert in the field. I was unable to find a second reviewer. However, the review I could collect is very detailed and, furthermore, I am myself quite familiar with the topic of this manuscript. Therefore, I feel confident making a decision with only one review. The review is positive but suggests several improvements. I agree with it. Therefore, I would like to invite you to revise your work following the reviewer's comments. Additionally, I would like to add two more comments. (i) it seems to me that you use reaction times as a proxy of emotionality. While this has been done several times in the past, more recent research has highlighted the fact that reaction time is more a measure of decision conflict than it is a measure of deliberation (Evans et al. 2015; Krajbich et al. 2015; Andrighetto et al. 2020). I think that you have to discuss this limitation of your work. (ii) The literature review regarding the role of emotion on moral judgments is quite incomplete. I have myself done some work on this, e.g., in Capraro, Everett & Earp (2019) we showed that priming emotion undermines instrumental harm, but not impartial beneficence. I also wrote a review on the topic that you might find helpful to find more references (Capraro, 2019)." Of course it is not a requirement to cite all these papers, but I am mentioning them, because they look related to your work.

Looking forward for the final revision.

References

Andrighetto, G., Capraro, V., Guido, A., & Szekely, A. (2020). Cooperation, Response Time, and Social Value Orientation: A Meta-Analysis. Proceedings of the Cognitive Science Society.

Capraro, V., Everett, J. A., & Earp, B. D. (2019). Priming intuition disfavors instrumental harm but not impartial beneficence. Journal of Experimental Social Psychology, 83, 142-149.

Capraro, V. (2019). The dual-process approach to human sociality: A review. Available at SSRN 3409146.

Evans, A. M., Dillon, K. D., & Rand, D. G. (2015). Fast but not intuitive, slow but not reflective: Decision conflict drives reaction times in social dilemmas. Journal of Experimental Psychology: General, 144(5), 951.

Krajbich, I., Bartling, B., Hare, T., & Fehr, E. (2015). Rethinking fast and slow based on a critique of reaction-time reverse inference. Nature communications, 6(1), 1-9.

---

## [Author Response · Author response to Decision Letter 1]

2 Feb 2021

Remark: Here, we only include the response to the second review round

Editor revision requests

Unfortunately, the reviewer who reviewed the first version of the manuscript is unable to review the revision. However, I have read the response letter and I am satisfied with the way you addressed their comments. However, I have noticed that you did not address the comments I gave on the original version of the manuscript, which I paste here.

We are sorry that we did not address those comments. We checked the e-mail dated November 5 2020 (PLOS ONE Decision: Revision required [PONE-D-20-30443] - EMID:7b51f57006a266e9) and it seems that the comments below were not included in the e-mail. We now have included changes in the manuscript based on those comments.

(…) (i) it seems to me that you use reaction times as a proxy of emotionality. While this has been done several times in the past, more recent research has highlighted the fact that reaction time is more a measure of decision conflict than it is a measure of deliberation (Evans et al. 2015; Krajbich et al. 2015; Andrighetto et al. 2020). I think that you have to discuss this limitation of your work. 

We thank the editor for indicating this important point. We clarify on page 9 that we do not con-sider longer reaction times as a measure of deliberation, but that our focus is indeed more on the aspect of decision conflict (this has also been emphasized again on page 20). To emphasize this further, we now consistently use the terminology of “emotional conflict” instead of “emotional engagement” throughout the paper when related to decision times. We also have clarified on page 3 that our considerations and hypotheses were not based on classic dual-process models (and we did not mention the dual-process approach in our original manuscript explicitly).

(ii) The literature review regarding the role of emotion on moral judgments is quite incomplete. I have myself done some work on this, e.g., in Capraro, Everett & Earp (2019) we showed that priming emotion undermines instrumental harm, but not impartial beneficence. I also wrote a re-view on the topic that you might find helpful to find more references (Capraro, 2019). Of course it is not a requirement to cite all these papers, but I am mentioning them, because they look related to your work.

We thank the editor for these additional insights and papers. We are indeed not arguing for a claim that cognitive and emotional components should be understood as antagonistic and we now have clarified this point in a remark on page 9 when introducing the reason measure. We also have included citations to this paragraph following the suggestions of the editor.

---

## [Editor Report · Decision Letter 2]

4 Feb 2021

Trolley Dilemma in the Sky: Context Matters When Civilians and Cadets Make Remotely Piloted Aircraft Decisions

PONE-D-20-30443R2

Dear Dr. Christen,

We’re pleased to inform you that your manuscript has been judged scientifically suitable for publication and will be formally accepted for publication once it meets all outstanding technical requirements.

Kind regards,

Valerio Capraro

Academic Editor

PLOS ONE
---

## [Editor Report · Acceptance letter]

10 Feb 2021

PONE-D-20-30443R2 

Trolley Dilemma in the Sky: Context Matters When Civilians and Cadets Make Remotely Piloted Aircraft Decisions 

Dear Dr. Christen:

I'm pleased to inform you that your manuscript has been deemed suitable for publication in PLOS ONE. Congratulations! Your manuscript is now with our production department. 

Kind regards, 

on behalf of

Dr. Valerio Capraro 

Academic Editor

PLOS ONE